# The Role of Plant Ubiquitin-like Modifiers in the Formation of Salt Stress Tolerance

**DOI:** 10.3390/plants13111468

**Published:** 2024-05-25

**Authors:** Siarhei A. Dabravolski, Stanislav V. Isayenkov

**Affiliations:** 1Department of Biotechnology Engineering, Braude Academic College of Engineering, Snunit 51, Karmiel 2161002, Israel; sergedobrowolski@gmail.com; 2International Research Centre for Environmental Membrane Biology, Foshan University, Foshan 528000, China; 3Institute of Agricultural and Nutritional Sciences, Martin Luther University Halle-Wittenberg, Betty-Heimann-Strasse 3, 06120 Halle, Germany; 4Department of Plant Food Products and Biofortification, Institute of Food Biotechnology and Genomics, The National Academy of Sciences of Ukraine, Baidi-Vyshneveckogo Str. 2a, 04123 Kyiv, Ukraine

**Keywords:** salt stress, ubiquitin, SUMOylation, autophagy, phytohormones

## Abstract

The climate-driven challenges facing Earth necessitate a comprehensive understanding of the mechanisms facilitating plant resilience to environmental stressors. This review delves into the crucial role of ubiquitin-like modifiers, particularly focusing on ATG8-mediated autophagy, in bolstering plant tolerance to salt stress. Synthesising recent research, we unveil the multifaceted contributions of ATG8 to plant adaptation mechanisms amidst salt stress conditions, including stomatal regulation, photosynthetic efficiency, osmotic adjustment, and antioxidant defence. Furthermore, we elucidate the interconnectedness of autophagy with key phytohormone signalling pathways, advocating for further exploration into their molecular mechanisms. Our findings underscore the significance of understanding molecular mechanisms underlying ubiquitin-based protein degradation systems and autophagy in salt stress tolerance, offering valuable insights for designing innovative strategies to improve crop productivity and ensure global food security amidst increasing soil salinisation. By harnessing the potential of autophagy and other molecular mechanisms, we can foster sustainable agricultural practices and develop stress-tolerant crops resilient to salt stress.

## 1. Ubiquitin-Mediated Protein Degradation in Plant Adaptation to Salt Stress

The climate of Earth is undergoing constant changes, which are often characterised by the increased number of frequent and severe adverse weather events such as inundations, hurricanes, drought, desertification, and others, making plants, more so than ever, dependent on effective mechanisms to withstand these environmental fluctuations [1,2]. In such challenging environmental conditions, for plants, given their stationary lifestyle, the effective removal and utilisation of malfunctioning and damaged proteins is crucial for the proper functioning of signal transduction systems and timely adaptation to environmental disturbances. Ubiquitination, a process of post-translational protein modification, plays a crucial role in almost every aspect of plant growth, development, and response to various stresses [3,4]. Thus, ubiquitin is the major component orchestrating protein degradation and turnover by ubiquitination [5]. The major ubiquitin-based protein degradation systems are ubiquitin (Ub)-26S proteasome system (UPS), small ubiquitin-like modifier (SUMO), and ubiquitin-like modifier autophagy-related protein 8 (ATG8) (Figure 1).

Salt stress is one of the most common adverse environmental factors affecting plant growth and productivity around the world. Improper irrigation practices, misuse of fertilisers, and global climate change in general gradually increase the severity of global soil salinisation [6]. Salt stress affects plant growth and productivity in various ways, such as osmotic and water stress, ion toxicity, nutrition disorders, dysregulation of biochemical, metabolic and physiological processes, membrane disorganisation, genotoxicity, repressed cell division, and others [7,8]. The stress-associated accumulation of reactive oxygen species (ROS) plays a crucial role in salt stress responses by causing oxidative damage to biomolecules and by initiating various cell signalling pathways [9]. Also, plants have developed several sophisticated defence mechanisms to alleviate salt stress, such as ion homeostasis, compartmentalisation and export, production of osmoprotectants and ROS scavengers, modulation of cytoskeleton dynamics and the cell wall composition, phytohormones-mediated reactions, and others [8,10]. Additionally, ROS accumulation is a crucial factor linking autophagy and salt stress responses [11]. The UPS and SUMO systems have been extensively reviewed in several recent reviews [12,13,14,15,16,17,18]; thus they will be introduced only briefly. Further, in this review we discuss the recent research deciphering the role of autophagy in salt stress tolerance, with a particular focus on ATG8 as the key player in selective autophagy.

## 2. Ubiquitin (Ub)-26S Proteasome System (UPS) and Its Role in Salinity Stress Tolerance

Ub is extremely conserved across eukaryotic kingdoms, with only minor amino acid variations between such distant organisms as humans and plants [19]. In the process of protein ubiquitylation, a 76-amino acid Ub polypeptide covalently attached to substrate lysine residues through the sequential activation by the E1 Ub-activating enzyme, resulting in conjugation onto an E2 Ub-conjugating enzyme, and subsequent catalysis by a specific E3 Ub ligase [20]. Such ubiquitylated proteins are recognised by the 26S proteasome for degradation [21,22]. Furthermore, the attachment of a single Ub (monoubiquitylation) can serve as a traffic signal for substrate transport within the cell or targeting for degradation in vacuoles or lysosomes, while the topology of polyubiquitylation can regulate protein activity and localisation, or target it for degradation by the 26S proteasome [23]. A diverse array of E3 ubiquitin ligases encoded in plant genomes plays the primary role in regulating the specificity of the UPS in targeting polyubiquitylated proteins for degradation [24]. Based on the structural compositions and the mechanics of the conjugation process of activated Ub moieties, plant E3 ligases can be classified into three main types: really interesting new gene (RING)-containing E3 ligases, homologous to the E6-AP carboxyl terminus (HECT), and plant U-box (PUB) (Figure 1) [25,26,27].

Considering specifically salt stress tolerance, UPS was recently shown to play a crucial role in various salt stress tolerance-related processes. Thus, the expression of the *Oryza sativa* (L.) *salt-induced RING Finger Protein 1* (*OsSIRP1*) gene was up-regulated in roots after salt, heat, drought, and ABA treatments. Furthermore, transgenic Arabidopsis plants expressing *OsSIRP1* showed lower salt stress tolerance during seed germination and root growth, suggesting that *OsSIRP1* acted through the ubiquitin 26S proteasome system to negatively regulate rice salt stress tolerance [28]. Another rice RING H2-type E3 ligase, OsSIRH2-14, was characterised as a positive regulator of salinity tolerance. Mechanically, OsSIRH2-14 interacted and regulated the stability of the HKT-type Na^+^ transporter OsHKT2;1 through the 26S proteasome system only in high salt but not in normal conditions. Accordingly, *OsSIRH2-14*-overexpressing rice plants showed a higher tolerance to salt stress and reduced Na^+^ accumulation in the aerial shoot and root tissues compared to wild-type plants [29]. Similarly, wheat (*Triticum*. *aestivum* L.) E3 ubiquitin ligase TaDIS1 (an orthologous of the *Oryza sativa* gene *drought-induced SINA protein 1*, a C3HC4 RING finger E3 ligase) was shown to interact and regulate the stability of the salt-tolerant protein (TaSTP) via the 26S proteasome pathway. Additionally, the expression level of TaSTP was up-regulated by salt, drought stresses, and ABA treatment, thus suggesting its involvement in different abiotic stress responses [30]. 

## 3. Small Ubiquitin Like-Modifiers (SUMO) and The Role in Salinity Stress Tolerance

Similarly to Ub, SUMO resulted in mono- or multi-SUMOylation of a substrate protein by attaching to the lysine residues (Figure 1) [31]. Interestingly, SUMOylation sites often have a consensus pattern (such hydrophobic/KxD/E motif, with x = any amino acid) recognised by a SUMO conjugation enzyme, while in some cases SUMO can be ligated onto its substrate even in the absence of a SUMO E3 ligase [18]. Recent proteome analysis demonstrated the significant enrichment of SUMOylated proteins involved in chromatin remodelling/repair, transcription, RNA metabolism, protein trafficking, and other vital functions [32,33], including also involvement in the response to various biotic and abiotic stresses [34,35,36,37]. However, it is important to note that SUMOylation and ubiquitylation intersect in multiple pathways and interact with various hormonal signalling pathways leading to antagonistic, synergistic, and even more complex outcomes [38]. In the case of abiotic stress tolerance, the ABA signalling pathway may be considered the primary target, which is modulated and regulated by Ub-SUMO systems [39,40,41].

Accordingly, the SUMO system was shown to play a crucial role in plant salt stress tolerance. Thus, overexpressing the rice Ubiquitin-like-specific protease 1D OVERLY TOLERANT TO SALT 1 (OsOTS1) increased the root biomass, reduced the levels of SUMOylated proteins, and increased salt stress tolerance. Also, salt treatment of rice plants with RNAi silenced the OsOTS1/2 increased accumulation of SUMO-conjugated proteins and greatly reduced salt stress tolerance, while in normal conditions these plants were not different from wild-type controls [42]. Similarly, the expression of rice SUMO E3 ligase SIZ1 (SAP AND MIZ1 DOMAIN CONTAINING LIGASE1) in Arabidopsis increased tolerance to heat, drought, and salt stresses and increased seed yields in stressed conditions. Also, *SIZ1-expressing* plants produced more osmoprotectant proline, accumulated more potassium ions, and excluded sodium ions more effectively in comparison to wild-type plants [43]. Furthermore, the heterologous expression of *SIZ1a/b/c* genes from sweet potato (*Ipomoea batatas* (L.) Lam.) in Arabidopsis *atsiz1* plants rescued the defective germination and growth phenotype, while *SIZ1a/b/c* expression in wild-type Arabidopsis plants enhanced tolerance to salt and drought stress [44]. Similarly, the expression of the SUMO-conjugating enzyme SCE1e from *Zea mays* (L.) in tobacco (*Nicotiana tabacum* L.) plants increased levels of SUMO conjugates and enhanced tolerances to salt and drought stress. Additionally, *ZmSCE1e*-expressing transgenic plants demonstrated a higher activity of antioxidant enzymes (SOD, CAT, APX, and GPX), lower H_2_O_2_ and malondialdehyde (MDA) accumulations, and higher expression of stress-responsive genes *LEA5*, *ERD10C*, *CDPK2*, and *AREB* under salt and drought stress treatments [45]. These results demonstrated that SUMO genes play a crucial role in salt stress tolerance by modulating protein sumoylation levels, antioxidant amount and activities, and the expression of general stress defence genes. 

## 4. Overview of the Core Plant Autophagy Components

Autophagy is a crucial regulatory system to maintain homeostasis; it is based on the degradation and recycling of damaged and malfunctioning cellular components during the normal plant’s life cycle and in response to environmental biotic and abiotic stresses [46]. Adverse environmental conditions like starvation, abiotic (such as salt, drought, cold, and heat) and biotic (such as pathogens) stresses are known to activate autophagy [47]. Depending on the target substrate and involved regulatory mechanisms, several forms of autophagy have been identified (macro-autophagy, micro-autophagy, organelle-specific, and chaperone-mediated autophagy) [48,49]. Despite some variation among plant species, different forms of autophagy have well-conserved core regulatory components identified in species ranging from lower photosynthetic algae to major agricultural species [50,51]. Originally, more than 40 *autophagy-related genes* (*ATGs*) were identified in yeast, while their homologues were further searched in *Arabidopsis thaliana* (L.) Heynh. and *Oryza sativa* L.—the major model species to study autophagy in plants. Among the core *ATG* identified in yeast, Arabidopsis has 3 *ATG1* homologues and 2 *ATG13*, while rice has 4 *ATG1* and 2 *ATG1* homologues. ATG11 acted as a scaffold protein to facilitate the formation of a kinase complex between ATG1 and ATG13 under starvation conditions [52]. 

Two sensor kinases identified in Arabidopsis and rice, TARGET OF RAPAMYCIN (TOR) and SUCROSE NONFERMENTING1-RELATED PROTEIN KINASE1 (SnRK1), have been found to cooperatively perceive nutritional status and to activate or inhibit autophagy by regulating ATG1–ATG13 kinase complex states [53,54]. Additionally, an ATG1-independent autophagy pathway has been identified in Arabidopsis under prolonged carbon starvation, where the SnRK1 catalytic SNF1 KINASE HOMOLOG 10 (KIN10) subunit directly phosphorylates the ATG6 subunit to trigger autophagy [55]. These data confirmed that, although the core autophagy mechanism is conserved in all eukaryotic cells, plants employ alternative pathways to activate autophagy to ensure an appropriate response to dynamic changes in the environment. 

The Arabidopsis homologue of yeast ATG14 was shown to regulate autophagosome assembly, while it was not yet discovered in rice [56]. ATG9 cycles between the ER and the cytoplasmic pool, regulating the trafficking of ATG18 on the autophagosomal membrane in a PI3P-dependent manner, thus ensuring efficient autophagosome budding from the initial autophagosome formed from the ER membrane [57,58]. 

Two essential ubiquitin-like conjugation systems, ATG8-PE (phosphatidylethanolamine) and ATG12-ATG5-ATG16, are crucial for the progression of autophagy. Thus, the formation of the ATG8-PE lipid complex requires cleavage by the cysteine protease ATG4, and the ATG12-ATG5 complex promotes ATG8 lipidation, while the specific function for ATG16 homologue in plants is still unknown [59,60,61]. ATG8 plays a central role in selective autophagy, because almost all currently known autophagy-related receptors/adaptors have an ATG8 interaction motif (AIM) or ubiquitin-interacting motif (UIM)-like sequences [62,63]. Subsequently, the autophagosome becomes laden with cargoes. The mature autophagosome subsequently fuses with and delivers its contents to the vacuole, where various hydrolases break down the contents into carbohydrates, amino acids, and lipids. These components are then returned to the cytosol and recycled to synthesize new products, or they are reused for other purposes.

### 4.1. The TOR-Dependent Autophagy Regulation and Its Interplay with Plant Phytohormones 

Through TOR kinase phytohormones, ROS, nutrients, amino acids, and various small molecules regulate the functions of ATG proteins and autophagy in plants [64]. Among amino acids, the role of cysteine has been extensively studied, with sulphur and carbon/nitrogen (C/N) precursors sensed by TOR kinase and GENERAL CONTROL NON-DEPRESSIBLE2 (GCN2) kinase, respectively. Further research demonstrated that elevated sulphur levels led to an increase in glucose levels, activating TOR, while low glucose levels during sulphur deficiency or impaired photosynthesis inhibit TOR and activate autophagy [65]. Prolonged carbon starvation and phosphate deficit resulted in autophagy through the induction of ER stress [55,66]. Nitrogen depletion inhibited TOR and activated autophagy; however, it also led to sugar accumulation (including glucose), which activated TOR and inhibited autophagy [67]. The deficit of many metals (such as zinc, iron, and manganese) also can induce autophagy, although the underlying mechanisms are poorly understood [68,69]. 

Phytohormones are crucial regulators of the TOR kinase and, subsequently, TOR-dependent autophagy (Figure 2). For example, **auxin** stimulated TOR activity thus inhibiting stress-induced autophagy [70]. Treatment with **salicylic acid** (**SA**) up-regulated autophagy during methyl jasmonate-induced leaf senescence [71]. Also, the accumulation of **SA** and ROS, associated with pathogens that attack and plant immunity, has been considered as autophagy-stimulating factors [72,73]. TOR inhibited **abscisic acid** (**ABA**) signalling under unstressed environmental conditions, while under stressed conditions ABA repressed TOR activity and activated autophagy [74]. The complex interplay between **brassinosteroids** (BRs) regulates plant growth, stress responses and nitrogen starvation. Thus, the blockage of BR signalling or biosynthesis up-regulated autophagy, while the increased activity of the BR pathway down-regulated autophagy. The brassinazole-resistant 1 (BZR1) transcription factor, brassinosteroid-insensitive 2 (BIN2) kinase, and regulatory-associated protein of TOR 1B (RAPTOR1B) have been defined as the major players coordinating interaction between BR and TOR [75,76]. The levels of active forms of **gibberellins** (GAs) and **cytokinin** (trans-zeatin) were lower in the anthers of autophagy-defective rice mutant *Osatg7–1*, while the exact role of these hormones in autophagy regulation remains unclear [77].

### 4.2. Plant Autophagy-Related Ubiquitin-like Modifier (ATG8) and Its Role in the Responses to Environmental Stresses 

ATG8 is the key component of the autophagy pathway, it conjugates with phospholipids to form a double-membrane vesicle (autophagosome), which further degrades intracellular components in selective or non-selective manners [78] (Figure 1). Unlike UPS, autophagosome has a much larger volume and is involved in bulk and selective recycling of huge protein complexes and aggregates, and malfunctioning and damaged organelles by delivering them to the vacuole. Since organelles (chloroplasts in particular) are extremely sensitive to various stresses, organelles’ recycling is considered as one of the primary targets in autophagy-mediated stress response regulation [reviewed in [79]]. Under normal conditions, the autophagy pathway engages at a basal level to maintain cellular homeostasis and facilitate plant development, but in the case of stress, it is further activated to provide necessary relief from a certain stress factor and to secure plant survival. Thus, ATG8 is tightly regulated by external and internal factors via plant hormones and diverse transcription factors, which, in total, contribute to plant survival during stress conditions [11,80].

ATG8, a protein crucial for the autophagic process, is normally activated through post-translational cleavage by the ATG4 protease at the C-terminus. ROS generated under stressed conditions inactivates ATG4 and guarantees a high level of ATG8 lipidation and, subsequently, autophagy progression. Additionally, the particular ATG4 homologues tend to interact with the preferred ATG8 isoforms, thus fine-tuning the specific and efficient induction of autophagy [59,81]. Furthermore, ATG8 can interact with several receptors and adaptor proteins, such as: (1)Neighbour of BRCA1 (NBR1) and the ATG8-interacting protein 1 (ATI1) and ATI2 proteins, which target plastid proteins for degradation upon carbon starvation [82];(2)Arabidopsis Orosomucoid (ORM) proteins 1 and 2, which are involved in plant immunity regulation through the degradation of Flagellin-Sensing 2 (FLS2) receptor kinase [83];(3)Tryptophan-rich sensory protein/translocator (TSPO), which is involved in the abiotic stress response through the regulation of the levels of PIP2;7 aquaporin on the cell surface [84];(4)DOMINANT SUPPRESSOR OF KAR 2 (DSK2), targeting the brassinosteroid pathway regulator BES1 for degradation [85];(5)A dehydrin, *Medicago truncatula (Gaertn.)* MtCAS31 (cold acclimation-specific 31), which under drought stress facilitated the autophagic degradation of MtPIP2;7, thus reducing water loss and improving drought tolerance [86];(6)S-nitrosoglutathione reductase 1 (GSNOR1) in hypoxic conditions [87];(7)ROOT HAIR DEFECTIVE3 (RHD3), an atlastin GTPases family member involved in ER stress response, was recently shown to have two AIM, through which it acted as a receptor for the selective autophagy of the ER (ER-phagy) during nutrient starvation. Thus, *rhd3* plants were defective in ER-phagy under ER stress. Furthermore, the RHD3–ATG8e interaction was enhanced under ER stress, but not *RHD3(ΔAIM1)* plants [88];(8)E3 enzyme Ufm1-protein Ligase 1 (Ufl1), one of the proteins regulating Ufmylation, a novel PTM characterised by multiple functions in diverse cellular processes, was shown to interact with ATG8e under salt stress. Furthermore, the core components in the Ufmylation cascade (Ufl1 and ubiquitin-fold modifier 1 (Ufm1)) interacted also with ATG1 and ATG6. Also, *Ufl1* plants showed salt hypersensitive phenotype and abnormal ER morphology, and the expression of Ufmylation cascade components was up-regulated by salt stress. These results suggested that autophagy machinery acted together with the Ufmylation cascade to maintain ER homeostasis under salt stress by regulating ER-phagy [89], and many others.

However, given the number and diversity of ATG8 isoforms, it is likely that particular ATG8s may be induced by different stress factors and interact with different effectors, while it is difficult to identify the role of each isoform because of the functional redundancy between isoforms and certain overlap in their inducers and interaction partners, which makes the analysis of single knock-out mutants ineffective [62,63,90,91].

### 4.3. Role of the Neighbour of BRCA1 (NBR1) Protein in the Regulation of Plant Stress Response 

NBR1, one of the best-characterised plant autophagy receptors, is a structural homologue and functional hybrid of mammalian autophagy receptors NBR1 and p62. In plants, NBR1 contains five domains (an N-terminal PB1 (Phox and Bem1p), a central ZZ-type zinc finger and NBR1_like, a two C-terminal UBA (ubiquitin-associated)), and an LC3-interacting region (LIR motif) (also known as AIM motif in yeast and plants) [92]. The biological functions of NBR1 have been investigated through an analysis of *nbr1* mutants or transgenic silencing lines. Arabidopsis *nbr1* plants showed normal growth and development under normal conditions and had no deviation in general and selective (peroxisomes, mitochondria, or the endoplasmic reticulum (ER)) autophagy [93]. Furthermore, NBR1 is not essential for age- and darkness-induced senescence but may modulate growth or senescence in short-day growth conditions or under mineral deficiency [94,95]. 

The role of NBR1 in plant abiotic stress tolerance is mediated by selective autophagy and, depending on its interaction with ATG8, it is also associated with the clearance of aggregation-prone misfolded proteins and protein aggregates [93]. Also, NBR1 was shown to participate in the regulation of plant tolerance to heat [96], oxidative [97], salt [98], and drought [99] stresses, and regulates heat stress memory [100] and resistance to some bacterial (like, for example, *Pseudomonas syringae* [101]) and viral (like cauliflower mosaic virus [102]) pathogens. In addition, Arabidopsis NBR1 was shown to interact with several crucial proteins of the ABA signalling pathway (such as ABSCISIC ACID INSENSITIVE (ABI3/4/5) [103], which suggested a wide role of autophagy in general stress-related processes. 

The role of NBR1 in salt stress tolerance is mediated through several mechanisms. As it was recently shown, the transgenic poplar hybrid plants’ (*Populus alba* × *P. tremula* var. *glandulosa*) overexpression *PagNBR1* had higher rates of enzyme activities of antioxidants (SOD, POD, and CAT) and a higher expression of ROS scavengers (*APX*, *MDHAR*, *DHAR*, and *GR*), resulting in lower malonaldehyde (MDA) concentration, electrolyte leakage, and membrane lipid peroxidation. Also, *PagNBR1* overexpression prevented a decrease in photosynthetic parameters (PSII conversion efficiency and maximum optical quantum yield), thus providing higher net photosynthesis rates under salt stress. Also, transgenic plants maintained a higher K^+^/Na^+^ ratio in the roots, which, at least partially, was mediated through the up-regulated expression of *SOS1*. Furthermore, PagNBR1 was localised in the autophagosome and interacted with ATG8, whose expression was also up-regulated in transgenic plants. Finally, transgenic plants formed more autophagosomes, resulting in a lower accumulation of insoluble ubiquitinated protein aggregates compared to wild-type plants under salt stress. In total, these results suggested that PagNBR1 enhanced salt tolerance by up-regulating antioxidant defence and ROS scavengers, protecting photosynthetic apparatus, maintaining the K^+^/Na^+^ ratio, and promoting the degradation of insoluble ubiquitinated protein in transgenic poplar plants under salt stress [104]. 

Therefore, NBR1-mediated selective autophagy plays a critical role in response and adaptation to both biotic- and abiotic stresses. Specific to the role of NBR1 in salt stress, it would be interesting to determine the possible regulatory function of NBR1 in other stress-related hormonal signalling pathways (such as ABA and SA), and in the degradation of ABA pathway regulatory proteins ABSCISIC ACID INSENSITIVE (ABI3, ABI4, and ABI5).

## 5. Universal Role of ATGs in the Regulation of Plant Salinity Stress

### 5.1. Stress Protective Role of ATGs in Microalgae

The unicellular alga *Chlamydomonas reinhardtii* (P.A.Dang.) has responded to salt stress through the activation of antioxidant enzymes, cell damage, autophagy, and nitric oxide metabolism processes. The application of comparative proteomics and physiological approaches helped to identify 74 proteins, such as antioxidant enzymes (Monodehydroascorbate reductase (MDHAR), Dehydroascorbate reductase (DHAR), Ascorbate peroxidase (APX), and glutathione reductase (GR)), nitric oxide metabolism-related proteins (S-Nitrosoglutathione reductase (GSNOR)), DNA damage (DNA damage checkpoint protein, DNA repair protein, DNA damage-inducible protein, RAN-binding proteins RANBP1, RAD 51, and REX1-B, and putative defender against death (DAD)) and autophagy (VPS30, ATP-dependent Clp protease proteolytic subunit, vacuolar trafficking protein, ATG3, and ATG8)), the accumulation of which was altered by short-term salt stress. These data suggested that ROS and GSNOR are involved in the activation of salt stress-induced autophagy and DNA damage [105]. Similarly, the cross-species meta-analysis on the transcriptomes of two microalgae (*Dunaliella salina* and *Dunaliella tertiolecta*) species under salt stress demonstrated that chaperone-mediated autophagy, lipid and nitrogen metabolism, photosynthesis, and ROS detoxification were core systems engaged in the *Dunaliella* salt stress response [106].

### 5.2. ATGs as Crucial Players in Adaptation to Various Abiotic Stresses

Recent genome-wide research has identified autophagy-related genes and their activation by different stresses in Foxtail millet (*Setaria italica* (L.) P. Beauvois), which is generally known for its strong resistance to various environmental stresses. From the total of 37 *SiATG* genes, the expression of 31 *SiATG* genes was induced by phytohormones (GA, methyl jasmonate (MeJA), SA, and ABA), 26 *SiATG* genes by cold, salt, and drought stresses, 24 *SiATG* genes by darkness, and 25 *SiATG* genes by nitrogen starvation [107]. In parallel, 49 *ATG* genes were identified in the genome of Tartary buckwheat (*Fagopyrum tataricum* (L.) Gaertn.). Among them, the eight *FtATG8* genes showed a high expression level under both salt and drought stresses. Also, all *FtATG8* genes contained MYB-family transcription factors binding and light-responsive cis-acting elements, and strongly correlated with other stress-associated transcription factors, suggesting their vital role in stress resistance in buckwheat plants [108]. 

In total, 29 *ATG* genes were identified in pepper (*Capsicum annuum* L.) another important agricultural species. Various abiotic stresses (heat, drought, cold, salt, and carbohydrate starvation) affected the expression of *CaATG* genes in a stress type-dependent pattern, thus suggesting the connection between autophagy and pepper tolerance to abiotic stresses. In particular, salt stress up-regulated eight *CaATGs* (*2*, *7*, *8a*, *8e*, *12*, *13a*, *18b*, and *18g*) and *VPS34*, with no *ATG* genes that were down-regulated. Stem salt stress up-regulated 15 *CaATG* genes (especially *5*, *8a*, *8c*, and *15*), while 4 *CaATG* genes were down-regulated (especially *2* and *4*) [109]. 

Also, nine well-expressed *ATG8* genes were identified in pear (*Pyrus* × *bretschneideri* Rehder). While *PbrATG8a/c/g/i* were up-regulated under salt stress conditions, only the *PbrATG8c* gene was up-regulated by all tested stresses (salt, drought, and *Botryosphaeria dothidea* infection) in pear [110]. Similarly, the expression of 35 *ATG* genes identified in sweet orange (*Citrus* × *sinensis* (L.) Osbeck) was affected by different stresses (salt, drought, cold, heat, and mannitol, and surplus manganese, copper, and cadmium). Under salt stress treatment, *CsATG8a*, *CsATG18e*, and *CsVTI12c* were up-regulated, while only *CsATG1a* was down-regulated in the seedlings. Furthermore, Arabidopsis plants transformed with *CsATG18a* and *CsATG18b* showed increased tolerance to salt, drought, cold, and osmotic stresses compared to control plants [111]. 

In potato (*Solanum tuberosum* L.), SA treatment, heat, and wounding stresses up-regulated the expression of *StATG3*, *StATG9*, *StATG11*, and *StATG13a*, while the expression of *StATG101* was up-regulated after wounding stress, but down-regulated by heat stress. Also, *StATG11* and *StATG101* expressions were up-regulated by salt stress. Furthermore, *StATG8* isoforms responded differently to various stresses, with *StATG8-2.1*, *StATG8-2.2*, *StATG8-3.2*, and *StATG8-4* up-regulated, while the expression of *StATG8-3.1* was down-regulated by heat stress, and *StATG8-1.1*, *StATG8-2.1*, and *StATG8-3.2* were up-regulated by wounding stress. Among the *StATG8* isoforms, only *StATG8-2.1* was up-regulated by SA treatment, and the expression of *StATG8-1.1*, *StATG8-2.1*, and *StATG8-3.2* was up-regulated after NaCl treatment [112]. 

In total, these results suggested that *ATG* genes are crucial players in response and adaptation to various abiotic stresses, which, at least partially, were mediated through interactions with different plant hormones. Further functional characterisations of *ATG* genes can provide promising new genetic resources for the development of novel stress-resistant varieties of crop species. 

### 5.3. Functional Characterisation of ATG8 Isoforms in Adaptation to Salt Stress

Experiments on marker GFP-ATG8a overexpressing lines demonstrated that autophagy is activated by salt stress, peaking at 30 min after stress exposure to alleviate the negative effects of salt stress on plants and to establish new homeostasis. In particular, autophagy removed more oxidised proteins, facilitated the accumulation of proline and soluble sugars, and promoted Na^+^ sequestration in the central vacuole of root cortex cells of the *ATG8* overexpressing plants. Such autophagy induction was absent in autophagy-defective plants (*atg2*, *atg5*, *atg7*, *atg9*, and *atg10*). Thus, *atg* mutants were hypersensitive to both osmotic and salt stress, while *ATG8* overexpressing plants showed stress-tolerant phenotypes, suggesting that a functional autophagy mechanism is absolutely required for salt stress tolerance in Arabidopsis [113].

In total, 77 *ATG* genes were defined in wheat (*Triticum aestivum* L.) and its di- and tetraploid-related species (51 in *T. turgidum* ssp. *dicoccoides* (wild emmer), 29 in *T. urartu*, and 30 in *Aegilops tauschii*). Further characterisations showed that transgenic wheat and Arabidopsis plants expressing *TaVAMP727* have higher resistance to cold, drought, and salt stresses [114]. Furthermore, 29 *ATG8* homologues were identified in allotetraploid rapeseed (AACC, *Brassica napus* L.). Among them, 13 *BnaATG8* genes were up-regulated in shoots, while 3 genes (*BnaA3.ATG8E*, *BnaA7.ATG8E*, and *BnaC3.ATG8E*) were down-regulated in the roots after salt treatment. Additionally, transgenic plants overexpressing the *BnaA8.ATG8F* gene showed reduced tolerance to salt stress, while the transcription level of *BnaA8.ATG8F* in the transgenic plants was about 30% lower in comparison to the wild type. Also, the total Na^+^ concentration was higher in the wild type than in *bnaa8.atg8f* plants, and the ratio of shoot/root Na^+^ accumulation in *bnaa8.atg8f* plants was almost three times that in wild type. These data suggested that under salt stress *BnaA8.ATG8F* expression is down-regulated and causes more Na^+^ translocation to shoots, thus reducing plant tolerance to salt [115].

In the poplar tree (*Populus*. *alba* × *P. tremula* var. *glandulosa*)), 48 *ATG* genes were identified and characterised. Thus, ABA treatment up-regulated *ATG1c/f*, *ATG2b*, *ATG4b*, *ATG5*, *ATG8a/g/j/k*, *ATG10*, and *ATG8a/c*, while *ATG1b/d*, *ATG2a*, *ATG3a/b*, *ATG7*, *ATG8c/d/e/f/h/i*, and *ATG13b/c/d* were down-regulated. The BR treatment, on the contrary, up-regulated *ATG7*, *ATG8a/b/j*, and *ATG18c/g/h*, while the *ATG1* subfamily was mostly down-regulated after BR treatment. Interestingly, in the *ATG18* subfamily salt treatment up-regulated mostly *ATG18a/c* members. Furthermore, *PagATG18a* overexpression enhanced salt stress tolerance and improved photosynthesis and antioxidant activities in salt-stressed poplar plants [116].

These results provide valuable information for further research on the functional characteristics of *ATG8* isoforms and other *ATG* genes in various agricultural and model species under biotic and abiotic stresses, interaction with hormonal signalling pathways and various stress response-related mechanisms, and prepare the necessary theoretical background for the future breeding of salt stress-tolerant varieties. 

## 6. Conclusions

In conclusion, in this review we discuss the role of ubiquitin-like modifiers in enhancing plant tolerance to salt stress, particularly focusing on the ATG8-mediated autophagy under salt stress. We have unveiled the multifaceted contributions of ATG8 to plant salt stress adaptation mechanisms. Specifically, ATG8 positively influences various physiological processes crucial for salt adaptation, including stomatal regulation, photosynthetic efficiency, osmotic adjustment, Na^+^ sequestration, antioxidant defence, and metabolic activities (Figure 3). Notably, *ATG8*-overexpressing plants exhibit enhanced photosynthetic capacity, improved osmotic adjustment, and superior growth performance compared to wild-type plants under salt stress. These results highlight the potential of autophagy modulation as a promising strategy for enhancing plant resilience to salt stress conditions, while minimising trade-offs between growth and stress tolerance. 

Moreover, we unveil the interconnectedness of autophagy with key phytohormone signalling pathways (such as abscisic and salicylic acids, brassinosteroids, and auxins). However, further exploration should focus on the molecular mechanisms underlying interplay with other hormones (such as cytokinins, and gibberellic and jasmonic acids), because their role in salt stress-stimulated autophagy is still unknown. Additionally, investigating the potential synergies between autophagy and other stress response pathways could unveil novel avenues for enhancing plant resilience to salt stress resistance. 

Overall, the discussed results suggested that understanding the molecular mechanisms underlying the role of ubiquitin-based protein degradation systems and particularly autophagy in salt stress tolerance may offer valuable insights for the design of innovative strategies to improve crop productivity and ensure global food security in the face of the globally increasing severity of soil salinisation. By harnessing the potential of autophagy and other molecular mechanisms, we can pave the way for sustainable agriculture practices that mitigate the adverse effects of salt stress on crop yields and develop novel stress-tolerant crops.

## Figures and Tables

**Figure 1 plants-13-01468-f001:**
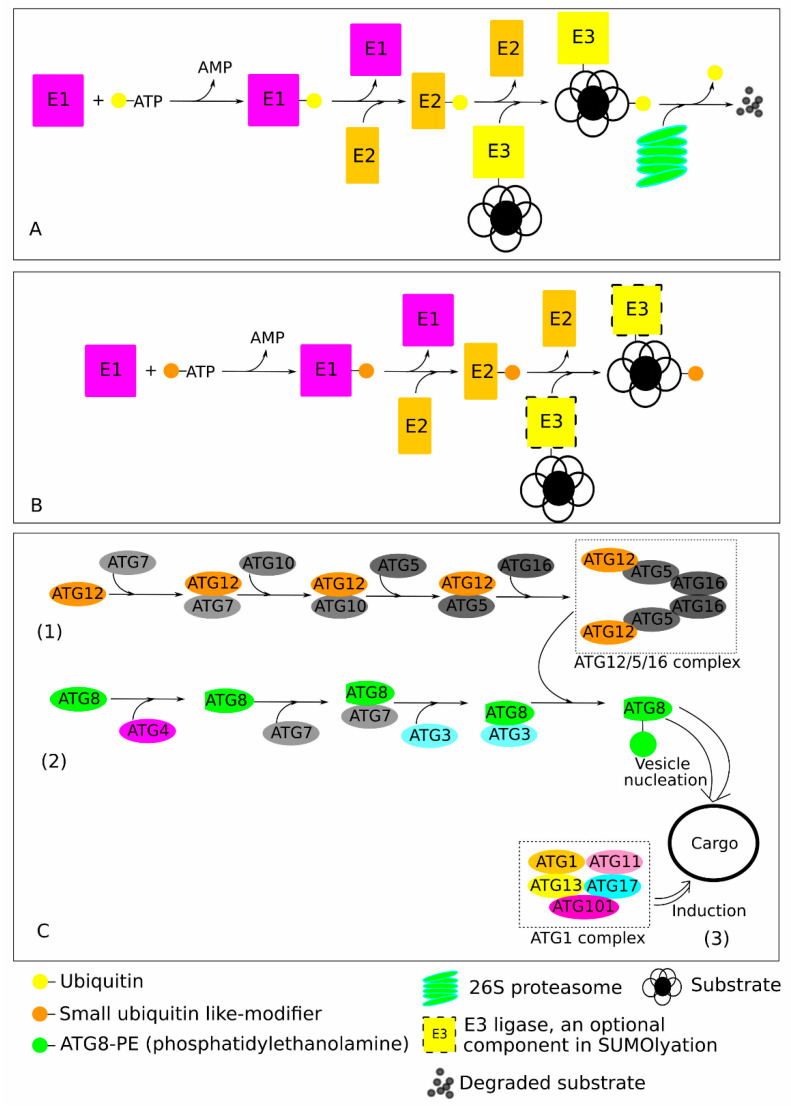
Schematic representation of the functioning mechanics of the ubiquitin (Ub), small ubiquitin-like-modifier (SUMO), and ubiquitin-like modifier autophagy-related protein 8 (ATG8). (**A**,**B**) The Ub and SUMOylation processes engaged similar cascade reactions with different enzymes (E1, E2, and E3) in an ATP-dependent way to add Ub or SUMO moieties, respectively, onto a target protein. In the UPS (**A**), marked proteins are recognised and degraded by the 26S proteasomes, which return back to the cytosol and the nucleus recycled Ub and short peptides. These short peptides then further degraded into free amino acids by various cellular peptidases. SUMOylation (**B**) may occur also in the absence of E3. (**C**) The autophagosome formation is mediated by various ATG proteins with three distinct processes: (1) formation of ATG12-ATG5-ATG16 complex; (2) sequential ATG8 cleavage and lipidation to form ATG8-PE (phosphatidylethanolamine); and (3) cargo, induced by the ATG1 complex, engulfed through interaction with ATG8 to form autophagosome.

**Figure 2 plants-13-01468-f002:**
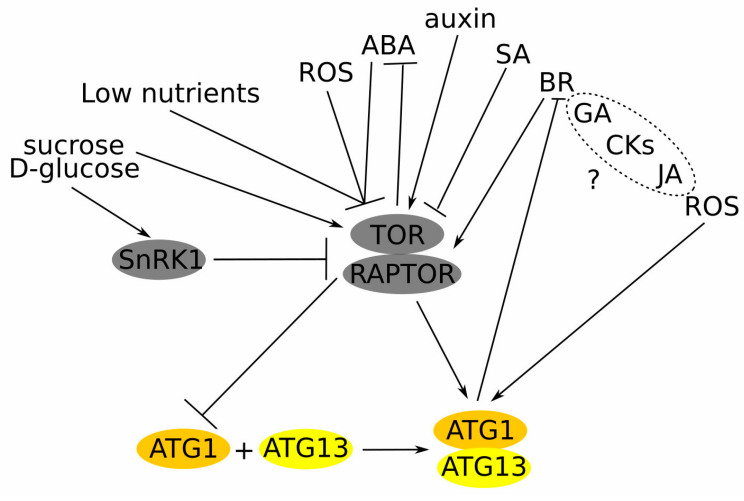
Autophagy sensors and regulators. The deficiency of nutrients activates autophagy through TOR inhibition. SnRK1 kinase senses sugars upstream of TOR. Glucose activates TOR, while glucose deficiency inhibits the TOR kinase, thus activating autophagy. ROS can activate autophagy by inhibiting TOR or acting directly through ATG downstream of the TOR kinase. ABA inhibits TOR kinase activity under stress, while in favourable conditions TOR suppresses ABA signalling. Auxin stimulates TOR, thus inhibiting the autophagy process. SA interacts with ROS to stimulate autophagy during senescence and plant immune response. BR inhibits autophagy through TOR activation, while autophagy regulates BR signalling by degrading its components. GA, CKs, and JA have been shown to regulate the autophagy process, although the exact mechanisms are still unknown (depicted with “?” and dashed circle). Arrows represent positive regulation and blunt lines—negative. TOR—target of rapamycin; RAPTOR—regulatory-associated protein of mTOR; SnRK1—sucrose non-fermenting related kinase 1; ROS—reactive oxygen species; ATG—autophagy related; ABA—abscisic acid; SA—salicylic acid; BR—brassinosteroids; CKs—cytokinins; GA—gibberellic acid; and JA—jasmonic acid.

**Figure 3 plants-13-01468-f003:**
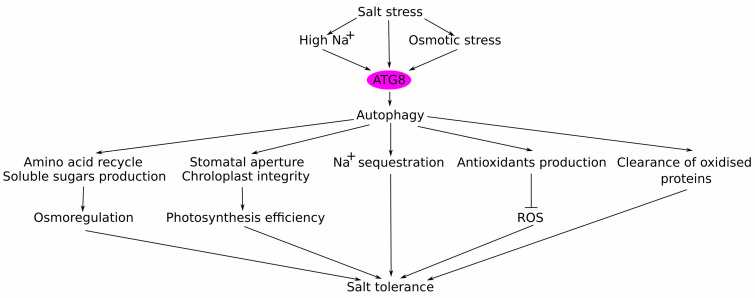
A proposed model on the regulatory roles of ATG8 and autophagy in salinity stress tolerance. There are five beneficial effects of autophagy: (1) soluble sugars’ production and amino acids’ recycling to improve plant osmotic status; (2) ATG8 contributes to the maintenance of optimal stomatal aperture and chloroplasts integrity, thus improving photosynthesis efficiency; (3) modulates expression/activity of various transporters, resulting in Na^+^ sequestration into the vacuole; (4) increased production of antioxidants, which reduces ROS levels and decreases oxidative stress; (5) salt stress-activated autophagy increases the clearance of oxidised and damaged proteins. In total, these beneficial effects increase plant tolerance to salt stress. Arrow lines indicate activation; blunt line—negative regulation.

## Data Availability

Not applicable.

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
