# Peer review of "The Role of Plant Ubiquitin-like Modifiers in the Formation of Salt Stress Tolerance"

_plants, 2024, doi:10.3390/plants13111468_

Round 1

Reviewer 1 Report

Comments and Suggestions for Authors

In this review, the authors discuss the role of ubiquitination in salt stress tolerance. After a brief introduction to the Ub-26S proteasome system and SUMO, which are discussed in detail elsewhere, the authors provide a detailed review of the currently available information on the role of autophagy in salt stress tolerance, with a particular focus on ATG8. The review is interesting and well written. In my opinion, it is definitely worth publishing in Plants. However, I have concerns that should be addressed before the paper can be accepted for publication.

Major points

1. I believe that better quality illustrations are needed nowadays.

2. In the review, there is a gap between the autophagy-related proteins discussed in detail and their effect leading to drought tolerance. It should be clarified how these proteins function to confer tolerance. It is expected that some proteins are selectively targeted to autophagy to activate tolerance, but the authors do not say anything about this crucial step. For example, the ATG8-interacting protein NBR1 is an adaptor protein that is thought to bind another protein and direct it to autophagosomes for degradation. What protein does NBR1 target for autophagy to induce drought tolerance? In this and other examples of such gaps, some explanation should be provided. In any case, even if no information is available, the steps between certain genes/proteins and their effect should not be ignored.

Minor points

Line 166. 4 ATG1 and 2 ATG1 homologues. Also, the numbers up to twelve should be given in words.

Line 199. Through TOR kinase - can be moved to the end of the sentence.

Line 253. "engages" should be "is engaged"

Line 314. dependents

Comments on the Quality of English Language

Minor editing of English language required

Author Response

Dear Editor and Reviewers,

We greatly appreciate your critical evaluation of our manuscript and helpful comments. Our reply to your comments would be provided point by point, where “A” stands for “Authors”, and “L” for “Lines”, where changes have been implemented. 

____________________________________________________________________________

In this review, the authors discuss the role of ubiquitination in salt stress tolerance. After a brief introduction to the Ub-26S proteasome system and SUMO, which are discussed in detail elsewhere, the authors provide a detailed review of the currently available information on the role of autophagy in salt stress tolerance, with a particular focus on ATG8. The review is interesting and well written. In my opinion, it is definitely worth publishing in Plants. However, I have concerns that should be addressed before the paper can be accepted for publication.

Major points

  1. I believe that better quality illustrations are needed nowadays.

A: We would like to believe that used schematics are sufficient to explain the required processes and involvement of particular players, therefore we do not want to make things more complicated. The quality of illustrations (dpi) fully corresponds to the requirements specified in the “Instructions for Authors” section.

  1. In the review, there is a gap between the autophagy-related proteins discussed in detail and their effect leading to drought tolerance. It should be clarified how these proteins function to confer tolerance. It is expected that some proteins are selectively targeted to autophagy to activate tolerance, but the authors do not say anything about this crucial step. For example, the ATG8-interacting protein NBR1 is an adaptor protein that is thought to bind another protein and direct it to autophagosomes for degradation. What protein does NBR1 target for autophagy to induce drought tolerance? In this and other examples of such gaps, some explanation should be provided. In any case, even if no information is available, the steps between certain genes/proteins and their effect should not be ignored.

A: Thank you, it is indeed a good question. Unfortunately, the majority of the current papers are descriptive in nature, where some features related to salt-tolerance/susceptibility are considered (Na+/K+ ratio and content, antioxidants expression and activity, photosynthetic apparatus performance, expression of various Na+ transporters, and so on) but provide no factual mechanical involvement of NBR1 or its interaction with specific targets. Some identified targets were listed in NBR1-focused reviews (please, see cited papers [62] and [93]) but, unfortunately, the information for salt tolerance is rather scarce. The manuscript was modified accordingly (please, see LL355-359).

Minor points

  1. Line 166. 4 ATG1 and 2 ATG1 homologues. Also, the numbers up to twelve should be given in words.

 A: Modified as suggested, Please see LL174-175

  1. Line 199. Through TOR kinase - can be moved to the end of the sentence.

A: Modified as suggested, Please see LL208-210

  1. Line 253. "engages" should be "is engaged"

A: Modified as suggested, Please see L262

  1. Line 314. dependents

A:  We have corrected the sentence.

  1. Comments on the Quality of English Language

Minor editing of English language required

A: The entire manuscript was further edited and proofread. 

Reviewer 2 Report

Comments and Suggestions for Authors

The abstract outlines a focused review on the role of ubiquitin-like modifiers, particularly ATG8-mediated autophagy, in plant salt stress tolerance. This specific focus is commendable as it allows for an in-depth exploration of a crucial mechanism contributing to stress resilience. However, some changes and modifications are required based on the following comments.

Earth or earth? correct it

Add some main findings and insights in the abstract.

After abstract provide a brief introduction and aims of the study.

The introduction should contain background of the study and need of the study.

Several E3 ubiquitin ligases have been shown to modulate the levels of key proteins in phytohormone signaling pathways like abscisic acid (ABA), which plays a crucial role in abiotic stress responses, including salt stress. Discuss these points in section 1.

Understanding the specific E3 ligases, their target substrates, and the associated molecular mechanisms involved in salt stress responses could provide opportunities for engineering salt stress tolerance in crops. Add here different approaches which can be used for this

“ Ubiquitination, a process of post-translational protein modification, plays a crucial role in almost every aspect of plant growth, development and response to various stresses” it would be better to add the mechanism.

Line 64-66 should be cited with recent studies. DOI: 10.3390/antiox12020268, DOI:10.1007/s10725-024-01128-y

Line 58-75 discuss reasons and implications of the salt stress in light of current climate changes.

The study provided mechanisms of different processes. However, some literature review is required like add table on Overview of the core plant autophagy components, with columns plants, stress, ATGs or hormone role in activation, autophagy components, types etc.

Section 4. Structural insights could further enhance the understanding of the molecular mechanisms involved in plant autophagy

Use consistent either italic or normal. Italic is standard for gene names

NBR1 Use consistent either italic or normal. Italic is standard for gene names

There are many gene names which must be italic.

Author Response

Dear Editor and Reviewers,

We greatly appreciate your critical evaluation of our manuscript and helpful comments. Our reply to your comments would be provided point by point, where “A” stands for “Authors”, and “L” for “Lines”, where changes have been implemented. 

____________________________________________________________________________

The abstract outlines a focused review on the role of ubiquitin-like modifiers, particularly ATG8-mediated autophagy, in plant salt stress tolerance. This specific focus is commendable as it allows for an in-depth exploration of a crucial mechanism contributing to stress resilience. However, some changes and modifications are required based on the following comments.

  1. Earth or earth? correct it

A: Yes, the planet, so capital Earth was used. 

  1. Add some main findings and insights in the abstract.
  2. After abstract provide a brief introduction and aims of the study.

The introduction should contain background of the study and need of the study.

A: The abstract was modified. The aim and background of our review are described in the last paragraph of chapter 1 (L77-80).

  1. Several E3 ubiquitin ligases have been shown to modulate the levels of key proteins in phytohormone signaling pathways like abscisic acid (ABA), which plays a crucial role in abiotic stress responses, including salt stress. Discuss these points in section 1.

A: We would like to point out that several E3 ubiquitin ligases relevant for salt stress tolerance have been discussed, please, see LL109-115 (OsSIRH2‐14), LL115-120 (TaDIS1), and LL142-151 (OsSIZ1). Therefore, We do not see it appropriate to repeat the same information again in Section 1.

  1. Understanding the specific E3 ligases, their target substrates, and the associated molecular mechanisms involved in salt stress responses could provide opportunities for engineering salt stress tolerance in crops. Add here different approaches which can be used for this

A: The information was added, please, see LL98-102.

  1. “Ubiquitination, a process of post-translational protein modification, plays a crucial role in almost every aspect of plant growth, development and response to various stresses” it would be better to add the mechanism.

A: The brief description of the mechanism was added (LL42-45).

  1. Line 64-66 should be cited with recent studies. DOI: 10.3390/antiox12020268, DOI:10.1007/s10725-024-01128-y

A: Unfortunately, the topics of the suggested two manuscripts from Fazal Ullah's Lab are not related to our manuscript:

- Khan M, Ali S, Al Azzawi TNI, Saqib S, Ullah F, Ayaz A, Zaman W. The Key Roles of ROS and RNS as a Signaling Molecule in Plant–Microbe Interactions. Antioxidants. 2023; 12(2):268. https://doi.org/10.3390/antiox12020268

and 

- Ullah, F., Saqib, S., Khan, W. et al. The multifaceted role of sodium nitroprusside in plants: crosstalk with phytohormones under normal and stressful conditions. Plant Growth Regul (2024). https://doi.org/10.1007/s10725-024-01128-y

Currently used citations [7,8] are more suitable and up to date. 

  1. Line 58-75 discuss reasons and implications of the salt stress in light of current climate changes.

A: Yes, indeed, among various abiotic stresses, drought and salt stress are considered as the major challenges for modern agriculture. 

  1. The study provided mechanisms of different processes. However, some literature review is required like add table on Overview of the core plant autophagy components, with columns plants, stress, ATGs or hormone role in activation, autophagy components, types etc.

A: Our manuscript is a literature review focused on the role of ubiquitin-like modifiers in salt stress tolerance. The analyses of all ATGs, various stresses, involved hormones etc. are far beyond the specified topic. Furthermore, the analysis of the autophagy components interaction with various hormones is usually conducted under normal (unstressed conditions). So far, to the best of our knowledge, we are not aware of studies deciphering the interaction between ATGs and different hormone signalling pathway components under various stresses. Otherwise, please, rephrase your comment so we could provide a proper reply. 

  1. Section 4. Structural insights could further enhance the understanding of the molecular mechanisms involved in plant autophagy

A: So far, the only solved crystal structure of proteins involved in plant autophagy is ATG12 (Please, see 10.4161/auto.1.2.1859 and 10.1248/bpb.b21-00439) which, however, are not relevant for the topic of our manuscript. The crystal structure of many human, yeast and other autophagy related proteins could be found in other reviews (please, see 10.3389/fcell.2020.00420).

  1. Use consistent either italic or normal. Italic is standard for gene names

A: Checked and modified throughout the entire manuscript.

  1. NBR1 Use consistent either italic or normal. Italic is standard for gene names

A: Checked and modified throughout the entire manuscript.

  1. There are many gene names which must be italic.

 A: Checked and modified throughout the entire manuscript.